# Robust Phase Association and Simultaneous Arrival Picking for Downhole Microseismic Data Using Constrained Dynamic Time Warping

**DOI:** 10.3390/s26010114

**Published:** 2025-12-24

**Authors:** Tuo Wang, Limin Li, Shanshi Wen, Yiran Lv, Zhichao Yu, Chuan He

**Affiliations:** 1State Key Laboratory of Lithospheric Evolution and Environment, Institute of Geology and Geophysics, Chinese Academy of Sciences, Beijing 100029, China; twang@mail.iggcas.ac.cn; 2Chongqing Xiangguosi Underground Gas Storage Co., Ltd., Chongqing 401121, China; lili_min@petrochina.com.cn; 3Shale Gas Institute of PetroChina Southwest Oil & Gasfield Company, Chengdu 610051, China; wenshanshi@petrochina.com.cn; 4Key Laboratory of Deep Petroleum Intelligent Exploration and Development, Institute of Geology and Geophysics, Chinese Academy of Sciences, Beijing 100029, China; ylpk8@mail.iggcas.ac.cn; 5School of Earth and Space Sciences, Peking University, Beijing 100871, China; chuanheus@163.com

**Keywords:** microseismic monitoring, downhole array, phase association, arrival picking, dynamic time warping

## Abstract

Accurate phase association and arrival time picking are pivotal for reliable microseismic event location and source characterization. However, the complexity of downhole microseismic wavefields, arising from heterogeneous subsurface structures, variable propagation paths, and ambient noise, poses significant challenges to conventional automatic picking methods, even when the signal-to-noise ratio (SNR) is moderate to high. Specifically, P-wave coda energy can obscure S-wave onsets analysis, and shear wave splitting can generate ambiguous arrivals. In this study, we propose a novel multi-channel arrival picking framework based on Constrained Dynamic Time Warping (CDTW) for phase identification and simultaneous P- and S-wave arrival estimation. The DTW algorithm aligns microseismic signals that may be out of sync due to differences in timing or wave velocity by warping the time axis to minimize cumulative distance. Time delay constraints are imposed to ensure physically plausible alignments and improve computational efficiency. Furthermore, we introduce a Multivariate CDTW approach to jointly process the three-component (3C) data, leveraging inter-component and inter-receiver arrival consistency across the entire downhole array. The method is validated against the Short-Term Average/Long-Term Average (STA/LTA) and waveform cross-correlation techniques using field data from a shale gas hydraulic fracturing. Results demonstrate that the proposed algorithm significantly enhances arrival time accuracy and inter-receiver consistency, particularly in scenarios involving P-wave coda interference and shear wave splitting.

## 1. Introduction

The development of unconventional hydrocarbon resources over the past two decades, driven primarily by breakthroughs in horizontal drilling and hydraulic fracturing technologies, has profoundly reshaped global energy supply dynamics [1,2,3]. In this context, microseismic monitoring has become an indispensable tool for unconventional hydrocarbon development, playing a pivotal role in the evaluation and optimization of hydraulic fracturing operations. By detecting and analyzing the small-scale seismic events induced during the high-pressure fluid injection, this technique enables real-time tracking of the fracture network propagation through the detection and analysis of the small-scale seismic events induced during high-pressure fluid injection [4]. Through the deployment of downhole or surface seismic arrays, operators can accurately reconstruct the spatio-temporal distribution of microseismic sources, thereby delineating the evolution of the Stimulated Reservoir Volume (SRV) and facilitating robust characterization of fracture geometry, connectivity, and treatment efficiency [5,6]. Accurate microseismic event characterization thus constitutes the cornerstone of reliable subsurface interpretation, and its precision is critical for informed decisions on well placement, fracturing design, and environmental risk management [7,8].

The accuracy of microseismic source location, focal mechanism estimation, and the subsurface velocity model reconstruction critically depends on two interdependent tasks: precise phase association (e.g., direct P-waves, direct S-waves, SH and SV waves from shear wave splitting, and occasionally reflected and refracted phases) and high-fidelity arrival time picking [9,10,11,12]. Conventional picking methods typically fall into two categories: (1) attribute-based approaches, which operate on individual receiver data and rely on handcrafted seismic attributes to detect abrupt changes in signal characteristics, with representative techniques including the Short-Term Average/Long-Term Average ratio (STA/LTA) [13], the Modified Energy Ratio (MER) [14], and kurtosis-based pickers [15]; and (2) correlation-based techniques that exploit waveform similarity across multiple receivers or between similar events to enhance detection robustness by identifying consistent arrival patterns [16,17]. More recently, data-driven artificial intelligence (AI) models, particularly deep learning architectures such as convolutional neural networks (CNNs) and recurrent neural networks (RNNs), have been introduced to automate the picking process. These models learn complex, nonlinear mappings between raw or preprocessed seismic waveforms and phase arrival labels from large annotated datasets, thereby capturing subtle patterns that are often missed by handcrafted features [18,19]. A key determinant of picking reliability is the signal-to-noise ratio (SNR) of the microseismic records [20]. Although moderate improvements can be achieved through preprocessing (e.g., denoising via wavelet transforms or singular spectrum analysis) or the integration of multi-attribute constraints (such as joint optimization of MER and the Akaike Information Criterion (AIC), or hybrid frameworks combining STA/LTA, polarization analysis, and AIC) [16,21], arrival time picking in downhole microseismic monitoring remains a significant challenge. Existing methodologies often struggle to reliably identify true first arrivals under complex field waveforms, such as near-source structural complexity, low inter-trace waveform coherence, and interference from overlapping seismic phases.

This challenge is particularly pronounced in specific downhole monitoring configurations, where conventional picking methods, such as time-windowed attribute analysis employing fixed thresholds or waveform cross-correlation for inter-trace time delay estimation under conditions of low waveform coherence, often yield inconsistent or erroneous picks, even for events exhibiting relatively high amplitudes. The core difficulty lies not in weak signal amplitude, but in the intrinsic complexity of the recorded wavefield. This complexity stems from three interrelated geophysical factors: (1) the close and geometrically heterogeneous proximity between receivers and the source in both lateral and vertical dimensions, (2) anisotropic radiation patterns dictated by the focal mechanism and fracture kinematics, and (3) pronounced velocity heterogeneity and sharp impedance contrasts within the near-source stratigraphic sequence. These conditions generate highly non-stationary, multi-path wavefields that severely distort first arrival signatures. Typical manifestations include P-wave coda energy obscuring the S-wave onset picking window and near-surface reflected phases arriving immediately after direct arrivals [10,22]. Consequently, conventional pickers struggle to reliably identify true onsets across a wide range of P/S energy ratios, ultimately introducing substantial errors into event location and source characterization [23].

To address the adverse impact of interfering preceding and trailing seismic phases (particularly P-wave coda obscuring S-wave onset windows and shear wave splitting producing multiple, closely spaced arrivals, as well as low inter-trace waveform similarity), this paper proposes a novel multi-channel arrival picking framework based on Constrained Dynamic Time Warping (CDTW). In contrast to conventional cross-correlation, which assumes linear time shifts between traces and is highly sensitive to waveform distortions, DTW aligns time series by identifying an optimal nonlinear warping path that minimizes the cumulative distance between two sequences [24]. Our approach explicitly incorporates two key physical priors intrinsic to downhole microseismic arrays: (1) waveform similarity across the receiver for a common source, despite distortion from near-source velocity heterogeneity and impedance contrasts, and (2) spatial smoothness of inter-receiver travel time differences, governed by local velocity structure and source–receiver geometry. By applying Multivariate CDTW to three-component (3C) downhole data, we reformulate the arrival picking and phase association problem as a robust multi-channel waveform synchronization task. This paradigm shift enables the simultaneous, coherent estimation of P- and S-wave first arrivals across the entire array, effectively suppressing ambiguities caused by phase overlap, coda interference, and split shear waves. The proposed method is validated using field microseismic data acquired during a hydraulic fracturing operation in a shale gas reservoir, demonstrating significant improvements in picking accuracy, inter-receiver consistency, and overall robustness.

## 2. Data and Methods

### 2.1. Data

The microseismic dataset analyzed in this study originates from a multi-stage hydraulic fracturing operation conducted in a horizontal shale gas well located in the Sichuan Basin, southwestern China. The operational target was the Wufeng–Longmaxi Formation, a marine shale unit characterized by high organic content and significant brittleness. The treatment well reached a total depth of approximately 2900 m, with a horizontal section extending over 1400 m. Microseismic monitoring was employed using a temporary sensor string installed in the inclined section of a nearby offset horizontal well. The array consisted of 12 levels of triaxial geophones with a nominal natural frequency of 15 Hz. Data were recorded with a sampling interval of 0.25 ms (i.e., a sampling rate of 4 kHz), and the spacing between adjacent geophone levels was set at 10 m. Notably, the average vertical distance between the perforation zones and the geophones was approximately 200 m, with monitoring distances ranging from 840 to 2162 m, as shown in Figure 1.

Figure 2 displays waveform records of two representative microseismic events recorded during distinct hydraulic fracturing stages (Stage 1 and Stage 10). Both events exhibit strong energy, with clear and coherent moveouts of the direct P- and S-wave arrivals across the array. However, low-frequency interference signals varying in both amplitude and duration are present on all geophone levels. These artifacts likely originate from near-wellbore noise or poor coupling between the geophones and the wellbore. Given the monitoring distances compared to the array aperture, the wavefront curvature is minimal across the array, resulting in approximately linear moveouts for direct P- and S-phases. Notably, significant coda energy following the direct P- and S-phases is observed in both events, which may result from the superposition of reflected phases generated by sharp impedance contrasts within the stratigraphic sequence. Specifically, the event in Figure 2a exhibits a relatively low P-wave signal-to-noise ratio, with a pronounced S-wave coda; the level of interference also varies across individual receivers. In contrast, the event in Figure 2b features a high-SNR P-wave, but its S-phase clearly manifests shear wave splitting. These distinct waveform characteristics arise from the interplay of multiple factors: the source mechanism of the microseismic events, the geometric configuration of the monitoring array, and the anisotropic properties of the subsurface formation. The variability in wavefield signatures, even within the same monitoring system, poses significant challenges for automated seismic phase identification and robust first arrival picking.

### 2.2. Dynamic Time Warping (DTW) and Its Multivariate Extension

Dynamic Time Warping (DTW) is a well-established algorithm for measuring similarity between two temporal sequences that may vary in speed, duration, or local timing distortions. Originally developed for speech recognition [25], DTW has since found broad applications in time series analysis, including seismic signal processing, where waveforms from the same source often exhibit non-uniform stretching or compression due to heterogeneous propagation paths, attenuation, and source–receiver effects [26,27,28,29,30].

Let two discrete microseismic traces recorded at different receivers be represented as time series:X = (*x*_1_, *x*_2_, …, *x*_n_) and Y = (*y*_1_, *y*_2_, …, *y_m_*),(1)
where *n* and *m* denote their respective lengths. DTW seeks an optimal warping path W = {(*i*_1_, *j*_1_), (*i*_2_, *j*_2_), …, (*i*_K_, *j*_K_)} through the *n* × *m* alignment grid that minimizes the cumulative distance between aligned samples, subject to three constraints:(1)Boundary conditions: the path starts at (1, 1) and ends at (*n*, *m*);(2)Monotonicity: indices *i_k_* and *j_k_* are non-decreasing (*i_k+_*_1_ ≥ *i_k_*, *j_k+_*_1_ ≥ *j_k_*);(3)Continuity: consecutive steps must be adjacent, typically restricted to the set {(0, 1), (1, 0), (1, 1)}.

The local distance between sample *x*_i_ and *y_j_* is typically defined as the squared Euclidean distance:d(*i*,*j*) = (*x*_i_ − *y_j_*)^2^.(2)

The cumulative distance *D*(*i*,*j*), representing the minimum total alignment cost to reach cell (*i*, *j*), is computed recursively via dynamic programming:(3)$$D(i,j)=d(i,j)+\min\left\{D(i-1,j),\;D(i,j-1),\;D(i-1,j-1)\right\},$$
with initial condition D(1, 1) = d(1, 1) and appropriate boundary handling for the first row and column.

The final DTW distance is given by D(*n*, *m*) and the optimal path W* is recovered by backtracking from (*n*, *m*) to (1, 1) through locally minimizing predecessors. This path provides a nonlinear, elastic alignment between two sequences, allowing corresponding features, such as first arrivals or phase onsets, to be matched even when distorted by wave superposition and dispersion.

In contrast to cross-correlation, which assumes a global constant time shift, DTW enables flexible, sample-wise alignment that accommodates local phase shifts and waveform broadening commonly observed in downhole microseismic data (e.g., the P- and S-wave coda interference in Figure 2a and the shear wave splitting in Figure 2b). Consequently, DTW can effectively synchronize waveforms that share a common source signature but exhibit morphological differences across receiver channels.

However, standard DTW operates on univariate time series and cannot directly exploit the full information content of three-component (3C) downhole geophone recordings. In practice, P- and S-wave energy is distributed across the X, Y, and Z components depending on source–receiver geometry, wave incidence angle, and local anisotropy. Critically, the SNR of a specific seismic phase can vary significantly among components, sometimes falling below the detection threshold on one or more channels, thereby hindering reliable arrival picking. To address this limitation, we adopt multivariate DTW (MDTW) [31,32,33], which generalizes the local distance to *C* dimensions (here, *C* = 3):(4)$$d_{mult}(i,j)=\sum_{c\in \{X,Y,Z\}}{\left(s_{i}^{\left(c\right)}-r_{j}^{\left(c\right)}\right)}^{2}$$
where $s_{i}^{\left(c\right)}$ is the amplitude of the source (reference) receiver on component *c* ∈ {X, Y, Z} at time index *i* and $r_{j}^{\left(c\right)}$ is the amplitude of the target receiver on the same component *c* at time index *j*. This formulation enables joint alignment of all three components, leveraging complementary phase information and enhancing robustness to component-wise SNR variability.

### 2.3. Constraint Dynamic Time Warping (CDTW)

Standard DTW allows unrestricted alignment between two time series, which can lead to physically implausible warping paths, particularly in the presence of noise, unrelated seismic phases, or long-duration recordings. To enforce temporal locality and ensure alignment consistency with expected wave propagation, this study imposes a global constraint on the warping path.

Specifically, we adopt the Sakoe–Chiba band constraint [34], which restricts the allowable alignment to a diagonal corridor of half-width around the main diagonal of the *n* × *m* cost matrix:(5)$$\left|i-j\right|\leq r$$
where *i* and *j* denote time samples in the reference and target sequences, respectively. The half-width *r* is typically set as a percentage (e.g., 10%) of the signal length, or more physically, is derived from the maximum expected inter-receiver travel time difference:(6)$$r=\Delta t_{\max}\cdot f_{s}$$
with $\Delta t_{\max}=L/v_{\min}$, where *L* is the inter-receiver spacing considered in the alignment, v_min_ is the minimum expected P- or S-wave velocity in the formation, and *f*_s_ is the sampling frequency.

This constraint ensures that only seismically coherent arrivals within physically plausible travel time bounds are aligned, effectively excluding spurious matches caused by ambient noise or late-arriving phases. Moreover, it reduces the computational complexity from O(*nm*) to O(*nr*), enabling scalable processing of large microseismic datasets. The resulting algorithm, referred to as Constrained DTW (CDTW), balances flexibility and physical realism. It retains DTW’s ability to accommodate nonlinear waveform distortions while preventing overfitting to noise or irrelevant signal components.

Figure 3 illustrates the application of Constrained Dynamic Time Warping (CDTW) to three-component microseismic data. Panel (a) displays the CDTW alignment and the corresponding optimal warping path between the x-, y-, and z-components (shown in blue, green, and red, respectively) of two microseismic signals recorded at geophone level 1 and 12 for the event depicted in Figure 2a. The imposed Sakoe–Chiba band constraint (200 samples in this example) restricts the warping path to physically plausible travel time differences, thereby avoiding spurious alignments with ambient noise or late-arriving phases. Panel (b) presents the resulting waveforms after elastic alignment: the same two three-component signals are now synchronized across all channels to maximize waveform coherence. This alignment effectively compensates for inter-channel travel time offsets and local waveform distortions, such as those caused by wave superposition and dispersion, enabling robust phase correlation.

### 2.4. Multivariate CDTW Framework for Downhole Array Data

Our complete M-CDTW processing workflow for downhole microseismic data is summarized in four key steps:
(1)Preprocessing and Coordinate Rotation

Raw 3C waveforms are first bandpass-filtered to suppress low-frequency and high-frequency noise. To ensure fair comparability across receivers while preserving the relative energy among components, this study applies amplitude normalization to preserve relative energy distribution among components while enabling fair inter-receiver comparison:(7)$$u_{k}^{\left(c\right)}(t)\leftarrow \frac{u_{k}^{\left(c\right)}(t)}{\max_{c^{\prime }\in \{X,Y,Z\}}\left(\max_{t}\left|u_{k}^{\left(c^{\prime }\right)}(t)\right|\right)},c\in \{X,Y,Z\},$$
where $u_{k}^{\left(c\right)}(t)$ denotes the waveform on component c at receiver k.

Crucially, due to the arbitrary azimuthal orientation of horizontal geophones in deviated or horizontal wells, we rotate the waveforms into a source–receiver-based radial–transverse–vertical (RTZ) coordinate system using the azimuth from receiver k to the perforation (as a proxy for the microseismic source location):(8)$$\left[\begin{array}{l} R_{k}(t)\\ T_{k}(t)\\ Z_{k}(t) \end{array}\right]=\left[\begin{array}{ccc} \cos\theta_{k} & \sin\theta_{k} & 0\\ -\sin\theta_{k} & \cos\theta_{k} & 0\\ 0 & 0 & 1 \end{array}\right]\left[\begin{array}{l} X_{k}(t)\\ Y_{k}(t)\\ Z_{k}(t) \end{array}\right],$$
where $\theta_{k}$ is the azimuth from receiver *k* to the perforation. This rotation concentrates P-wave energy in the Z component and S-wave energy primarily in the R and T components, enhancing inter-receiver waveform coherence.

(2)Reference Trace Selection and Initial Picking

A reference receiver *R* is selected based on the highest SNR or the earliest P-wave arrival. P- and S-wave onsets on the reference receiver are manually or automatically picked using a robust single-receiver picker (e.g., STA/LTA or MER). These picks serve as temporal anchors for subsequent multi-receiver alignment.

(3)Time Difference Constraint Definition

As described in Section 2.3, the Sakoe–Chiba band half-width *r_k_* is determined based on the maximum expected inter-receiver travel time difference, which accounts for the array geometry and formation velocity model. Specifically, for alignment between the reference receiver and a target receiver k, we set the following:(9)$$r_{k}=\frac{L_{k}}{v_{\min}}f_{s},$$
where $L_{k}$ is the inter-receiver distance between the reference and receiver *k*, v_min_ is the minimum expected P- or S-wave velocity in the formation, and *f*_s_ is the sampling rate (in Hz). This adaptive constraint ensures that the warping path remains causally and physically meaningful for each receiver pair.

(4)Pairwise M-CDTW Alignment and Robust Time Difference Estimation

For each non-reference receiver *k*, we perform M-CDTW alignment between the 3C waveform and the reference traces. Rather than estimating the relative arrival time from a single point on the warping path, which is sensitive to local noise, we compute a robust average time shift over a short coherent segment surrounding the expected phase onset. Specifically, for the P-wave, we select *N* consecutive point pairs ${\left\{(i_{l},j_{l})\right\}}_{l=1}^{N}$ within a temporal window spanning approximately one to two effective wavelengths and calculate the mean time delay:(10)$$\Delta t_{k}^{P}=\frac{1}{N}\sum_{l=1}^{N}\left(j_{l}-i_{l}\right)/f_{f},$$

A similar procedure is applied for the S-wave using a window after the reference S-pick $t_{S}$. The absolute arrival times at receiver k are then estimated as follows:(11)$$t_{k}^{P}=t_{rf}^{P}+\Delta t_{k}^{P},t_{k}^{S}=t_{rf}^{S}+\Delta t_{k}^{S},$$

This framework simultaneously achieves phase association, arrival picking, and noise-robust synchronization across the entire array, leveraging both inter-component and inter-receiver consistency. As a result, it enables accurate microseismic event characterization even under challenging conditions such as the effects of waveform superposition and dispersion.

## 3. Performance Evaluation

This study demonstrates the feasibility of the proposed methodology through the processing and analysis of two representative microseismic events. For each hydraulic fracturing stage, we first utilized the perforation shot signal to determine the azimuthal orientation of the downhole three-component (3C) geophones. This enabled rotation of the recorded waveforms into a consistent geographic coordinate system (e.g., North–East–Vertical, NEV), ensuring uniform component definitions across the array, thereby ensuring consistent component definitions across the array. To enhance signal quality, all recordings were treated with a zero-phase Butterworth bandpass filter (50–350 Hz), effectively attenuating low-frequency drift and high-frequency ambient noise while preserving waveform phase integrity. Among all receivers, the 12th receiver, which recorded the earliest first arrivals for both events, was selected as the reference for initial arrival time picking. To balance computational efficiency and workflow simplicity, we adopted the widely used short-term average/long-term average (STA/LTA) ratio method for automatic P- and S-wave onset detection. The algorithm employed a short time window of 100 samples, a long time window of 500 samples, and a trigger threshold of 5. The resulting P- and S-wave picks on the reference trace are shown in Figure 4 and serve as temporal anchors for subsequent multi-receiver alignment within the M-CDTW framework.

We employed the M-CDTW algorithm between the reference receiver and all other receivers to achieve robust waveform alignment via nonlinear stretching. To ensure physically plausible warping paths, we adopted a dynamic time difference constraint that adapts to the inter-receiver geometry, as defined in Equation (9). In the specific case shown in Figure 5, comparing the reference receiver with the seventh receiver, this constraint evaluates to 80 samples, computed using a minimum S-wave velocity of 2000 m/s, an inter-receiver spacing of 40 m, and a sampling rate of 4000 Hz. Figure 5 demonstrates that, by jointly utilizing all three components, the M-CDTW successfully aligns both P- and S-wave phases across all channels. The method reliably handles complex arrivals such as shear wave splitting (e.g., the SH wave in Figure 5d and the SV wave in Figure 5b). To accurately determine the inter-receiver time delays, we focused on short temporal windows after the P- and S-wave onsets were identified on the reference trace. These windows, highlighted by red (80 samples) and green (120 samples) line regions in Figure 5, capture the most coherent portion of the reference phase while minimizing contamination from adjacent arrivals or noise. Within each window, the sample indices along the optimal warping path were averaged to estimate the corresponding (unwarped) sample location in the target trace. This averaged mapping provides a stable estimate of the relative travel time shift between adjacent receivers, which is then used to derive the first arrival times for all receivers, as subsequently illustrated in Figure 6.

Typically, time-corrected (flattened) records provide an intuitive and effective way to validate the accuracy and consistency of first arrival picks [17]. Taking the two events in Figure 6 as examples, their first arrival picking results are shown in Figure 7. When the waveforms are shifted according to these picks (i.e., aligned to a common reference time), the resulting flattened sections exhibit highly coherent P- and S-wave phases across the array, confirming the reliability of the arrival time estimates.

To elucidate both the limitations of conventional first arrival picking methods in complex microseismic wavefields and the underlying causes, we analyze the representative event in Figure 2a using two widely adopted approaches: the short-term average/long-term average (STA/LTA) ratio method and cross-correlation-based time delay estimation. In time-windowed attribute-based picker such as STA/LTA, the selection of analysis windows and picking criteria plays a critical role in determining the final results. However, window design is often subjective and lacks universal optimality [14,15,16]. This subjectivity becomes particularly problematic when processing large-scale microseismic datasets, where phase characteristics may vary significantly across receivers and events. For the STA/LTA method, we tested two sets of arrival picking parameters. The first set matched those used in prior analyses (short window = 100 samples, long window = 500 samples, trigger threshold = 5), while the second set employed a short window (50 samples), a long window (300 samples), and a lower trigger threshold (3), a configuration designed to enhance sensitivity to subtle onsets. Figure 8 shows the STA/LTA energy ratios and corresponding first arrival picking results for all geophones using the first set of parameters. When the STA/LTA ratio did not exceed the threshold, we conservatively selected the nearest local maximum as the first arrival pick, as illustrated for the sixth receiver in Figure 8. This is primarily due to persistent noise interference, which prevents accurate phase picking under this parameter set.

The time-corrected (flattened) first arrival results obtained with both parameter sets are compared in Figure 9. As evident from the figure, the choice of parameters significantly affects the picking accuracy. Minor adjustments to window lengths or thresholds can lead to inconsistent or physically implausible travel time estimates across the array. These observations underscore that optimal parameter selection must be carefully tailored to the specific characteristics of each microseismic event, such as frequency content, signal-to-noise ratio, and phase complexity. In contrast to STA/LTA methods that rely on manually tuned parameters across receivers, our proposed approach eliminates the need for empirical window or threshold specification. By leveraging multi-component waveform coherence through M-CDTW alignment, it achieves objective, consistent, and physically constrained arrival time estimates, thereby circumventing the subjectivity and instability inherent in conventional STA/LTA-type techniques.

The cross-correlation method, while a widely used technique for estimating seismic time delays between adjacent receivers, relies fundamentally on waveform similarity [16,17]. Conventionally, the time delay is determined by identifying the time lag corresponding to the peak of the normalized cross-correlation function. However, dispersion and coda interference lead to reduced waveform similarity between the 12th (black) and 7th (blue) traces (Figure 10a), producing a flat and multi-peaked cross-correlation function that complicates reliable time delay estimation (Figure 10c). When the global maximum is used to estimate the time delay under such conditions, it leads to substantial waveform misalignment, as shown in Figure 10e. This limitation also adversely affects S-wave alignment. Even when S-wave cross-correlation peaks appear relatively distinct, interference from residual coda, noise, or overlapping phases can shift or distort the true peak location, thereby directly degrading the accuracy of first arrival picks (Figure 11). Although hybrid strategies, such as integrating inter-receiver time delay consistency constraints, can mitigate erroneous picks through post-processing, they often require ad hoc tuning and may not generalize well across varying datasets. By accounting for nonlinear temporal distortions and leveraging multi-component signal redundancy, MCDTW yields significantly more consistent and physically plausible first arrival estimates compared to standard cross-correlation.

## 4. Discussion

To systematically evaluate the reliability of the proposed method, we applied it to direct P- and S-wave first arrival picking for 1115 microseismic events identified from monitoring data across 19 hydraulic fracturing stages, and compared its performance against a conventional multi-trace cross-correlation (MTCC) automatic picking approach [16]. Following the same processing workflow described earlier, we selected the 12th receiver as the reference trace and employed the STA/LTA method, with a short window of 100 samples, a long window of 500 samples, and a trigger threshold of 5 (as previously defined), to obtain an initial arrival estimate. Based on this preliminary pick, a suitable time window (300 samples) was extracted to compute pairwise inter-receiver differential travel times, which were then used to implement the MTCC-based refinement procedure.

The performance of both methods was quantitatively assessed using two key metrics: (1) the concentration (compactness) of inter-receiver differential travel time distributions, and (2) the multi-trace similarity coefficient computed after arrival time correction, both of which characterize the spatial consistency and physical plausibility of the picked arrivals. Given the compact geometry of the source region relative to the downhole receiver array (i.e., the monitoring aperture is much smaller than the typical source–receiver distance), spatially proximal microseismic events are expected to exhibit highly consistent travel time characteristics: specifically, (i) the P-S arrival time difference at a given receiver, which reflects source–receiver distance, should be similar across neighboring events, and (ii) the inter-receiver time differences for the same seismic phase (P- or S-wave), which reflect path-length variations, should follow a regular pattern, such as approximately equal time increments between adjacent receivers. Consequently, significant deviations from this expected behavior serve as a robust indicator of either systematic or random picking errors.

Figure 12 presents the distributions of two representative differential time types obtained from both methods:(1) the travel time difference between the first and last receivers for P- and S-wave phases, and (2) the P-S time lag at the reference receiver. The results show that the MTCC method (blue) yields notably scattered distributions, particularly in the inter-receiver delays of coherent phases, with wide and irregular spread. In contrast, the proposed M-CDTW method (red) produces significantly more concentrated distributions, indicating more accurate internal timing relationships among events and receivers. Although truth arrival times are unavailable for direct validation, the markedly reduced dispersion strongly suggests that our method achieves lower picking uncertainty, thereby providing more reliable temporal constraints for subsequent event location.

Furthermore, the quality of first arrival picks can be independently validated using a multi-trace similarity coefficient computed after arrival time correction. This metric is obtained by aligning all traces according to the picked arrivals to minimize residual inter-receiver time shifts and then quantifying the resulting waveform coherence. Specifically, the multi-trace similarity coefficient is defined as the average zero-lag normalized cross-correlation across all unique receiver pairs following alignment, serving as a robust proxy for overall waveform consistency. As shown in Figure 13, panels (a) and (b) display the similarity coefficients for P- and S-wave phases, respectively. A clear improvement is observed: the proposed method consistently achieves higher similarity coefficients across all events, demonstrating superior waveform alignment accuracy and stronger cross-trace coherence for both P and S phases.

Additionally, Figure 14 displays fixed-length time windows extracted around the P-wave arrivals picked by our method, sorted in descending order of P-S time differences (i.e., from farthest to nearest source–receiver distances). The S-wave phase clearly exhibits a continuous and smoothly varying moveout trend, visually confirming the high accuracy of the majority of first arrival picks. Notably, within the central fracturing stages, a clear shear wave splitting phenomenon is clearly observable in the recordings. This feature not only aligns with expected anisotropic behavior in the fractured reservoir but further corroborates the fidelity of the picked onsets, as such subtle wavefield characteristics would be obscured or distorted by inaccurate timing. Nevertheless, it should be acknowledged that under low signal-to-noise ratio (SNR) conditions, the absolute pick on the reference trace may still contain non-negligible errors. Since the differential time correction and multi-trace alignment procedures rely on this initial reference pick, any error therein can propagate and indirectly degrade the alignment accuracy of other traces. This limitation fundamentally stems from the poor distinguishability between weak microseismic signals and ambient background noise in the time domain on individual receivers, particularly for events with low energy or those recorded at greater distances. Despite this inherent challenge, the overall coherence and structural continuity observed in Figure 14 underscore the robustness of the proposed picking strategy in practical monitoring scenarios.

Admittedly, no single picking method can achieve ideal performance across all complex real-world scenarios. Nevertheless, the M-CDTW framework adopted in this study effectively mitigates the strong reliance of conventional approaches on waveform similarity by jointly leveraging multi-component waveform redundancy and nonlinear time-warping capabilities. As demonstrated in the results, the proposed method exhibits clear advantages over traditional techniques, particularly in handling waveform superposition and multiply phase interference. That said, the discussion above also reveals that the current implementation still has considerable room for improvement to better accommodate datasets with diverse characteristics. For instance, under extremely low signal-to-noise ratio (SNR) conditions or in geologically complex settings, further refinement, such as adaptive parameter tuning or incorporation of additional physical or statistical constraints, may be necessary to enhance picking accuracy. Moreover, for specific types of microseismic events (e.g., very shallow or deep events), algorithmic adjustments may be required to improve robustness and generalizability.

## 5. Conclusions

This study presents a robust, multi-channel arrival picking framework for downhole microseismic data based on Multivariate Constrained Dynamic Time Warping (M-CDTW). By reformulating phase association and first arrival estimation as a joint waveform synchronization problem, our method overcomes critical limitations of conventional single-receiver or cross-correlation-based approaches, particularly in complex wavefields typical of hydraulic fracturing operations.

Unlike traditional techniques that process P- and S-wave picks independently, M-CDTW simultaneously estimates both phases while enforcing inter-receiver and inter-component consistency. This holistic strategy not only improves individual pick accuracy but also ensures expected geometric moveout patterns, providing more reliable constraints for subsequent event location and source inversion.

Validation using a field dataset from a shale gas reservoir in the Sichuan Basin demonstrates that the proposed method achieves substantially higher picking accuracy and inter-receiver consistency compared to standard multi-trace cross-correlation (MTCC) approaches. This is evidenced by more concentrated differential travel time distributions, elevated multi-trace similarity coefficients after alignment, and coherent S-wave moveout trends—including the clear resolution of shear wave splitting in central fracture stages. Although challenges remain under extremely noisy conditions due to reference pick uncertainty, the overall performance underscores the method’s reliability and objectivity.

## Figures and Tables

**Figure 1 sensors-26-00114-f001:**
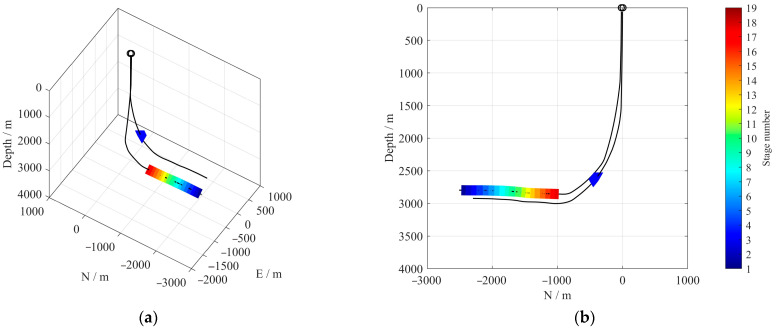
(**a**) Three-dimensional view schematic of the monitoring array deployment; (**b**) vertical cross-section along the YZ plane. Blue triangles denote the positions of the geophones, colored rectangular zones indicate different hydraulic fracturing stages, and black lines represent the well trajectories.

**Figure 2 sensors-26-00114-f002:**
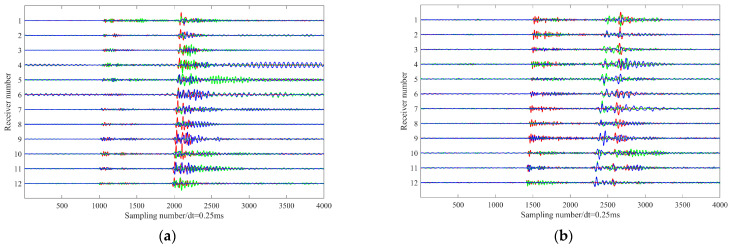
(**a**) Waveform of a microseismic event recorded during Stage 1 of the hydraulic fracturing treatment; (**b**) Waveform record of a microseismic event recorded during Stage 10 of the hydraulic fracturing treatment. The blue, green, and red lines represent x-, y-, and z-components seismograms, respectively.

**Figure 3 sensors-26-00114-f003:**
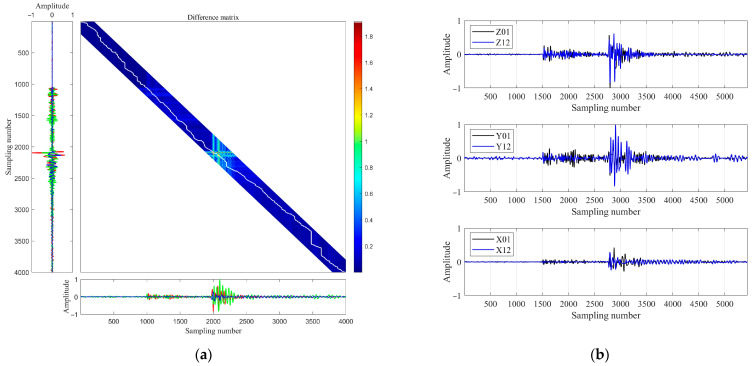
(**a**) Constrained Dynamic Time Warping (CDTW) alignment and optimal warping path between the x- (blue), y- (green), and z- (red) components of two microseismic signals recorded at geophone levels 1 and 12 (event from Figure 2a). The Sakoe–Chiba band constraint (200 samples) enforces physically plausible alignments. The white line represents the optimal warping path; (**b**) waveforms after CDTW-based elastic alignment, showing synchronized three-component signals with enhanced coherence.

**Figure 4 sensors-26-00114-f004:**
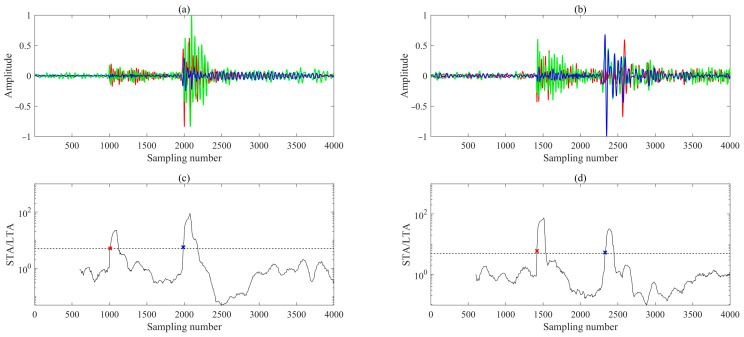
(**a**,**b**) represent three-component waveforms recorded by the 12th receiver of the event shown in Figure 2a and Figure 2b, respectively. (**c**,**d**) Corresponding STA/LTA ratios for the records in (**a**,**b**), respectively. The blue, green, and red lines represent N-, E-, and V-components seismograms, respectively. The dashed horizontal line indicates the STA/LTA trigger threshold for arrival detection; red and blue ‘×’ marks denote the picked P- and S-wave first arrivals, respectively.

**Figure 5 sensors-26-00114-f005:**
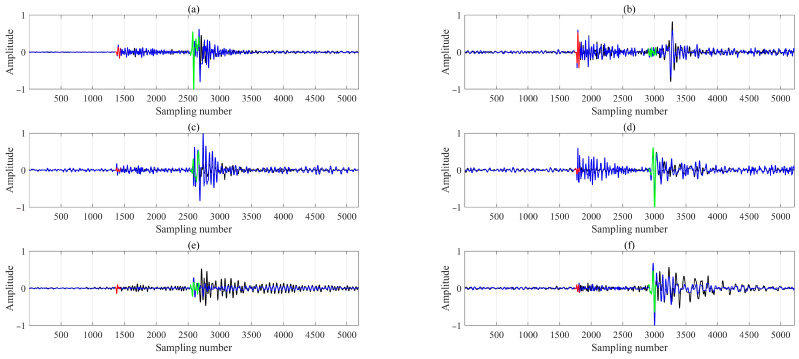
M-CDTW-based waveform alignment between the reference receiver (black) and a target receiver (blue) for two events. Panels (**a**,**b**), (**c**,**d**), and (**e**,**f**) correspond to the three rotated components. Red and green lines (80 and 120 samples, respectively) mark the coherent segments used to estimate inter-receiver travel time delays.

**Figure 6 sensors-26-00114-f006:**
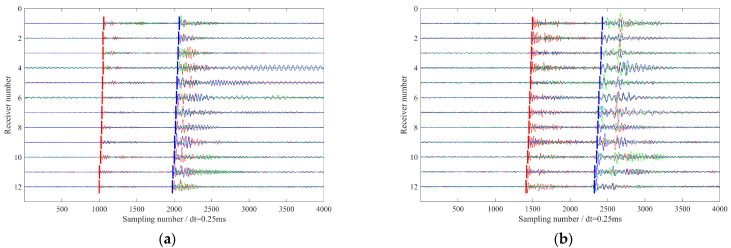
(**a**) First-arrival picks for the microseismic event from Stage 1; (**b**) First-arrival picks for the microseismic event from Stage 10. Red and blue vertical lines indicate the P- and S-wave first arrival picks, respectively. The blue, green, and red traces represent the N-, E-, and V-components seismograms, respectively.

**Figure 7 sensors-26-00114-f007:**
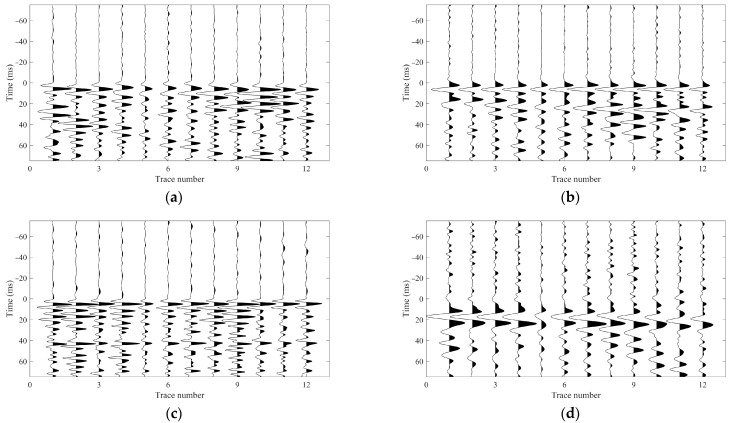
Flattened waveforms for the two microseismic events in Figure 6, obtained by aligning all receiver traces to the picked P- and S-wave first arrivals. Panels (**a**,**b**) correspond to the P- and S-wave alignments for the event in Figure 6a, and panels (**c**,**d**) to those for the event in Figure 6b.

**Figure 8 sensors-26-00114-f008:**
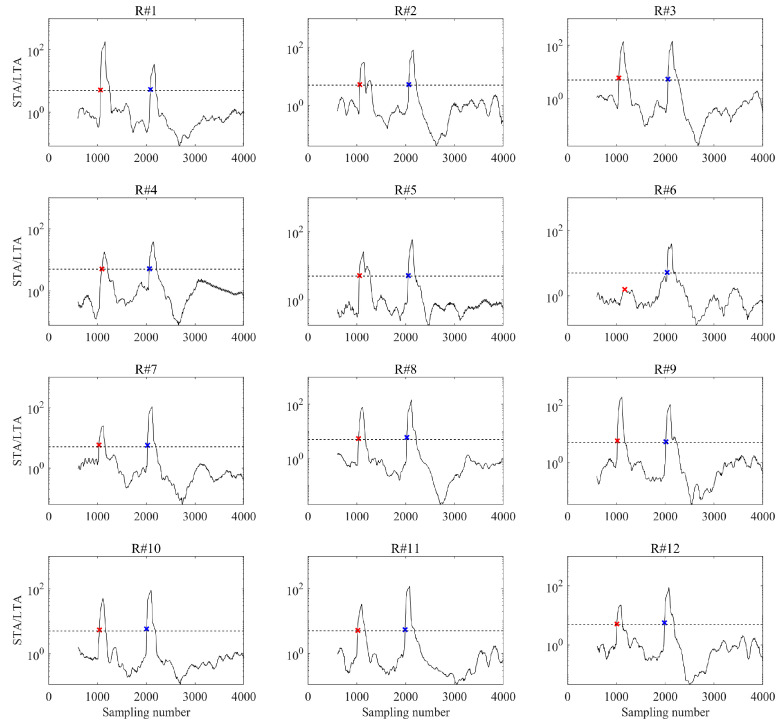
STA/LTA energy ratios and corresponding P- and S-wave first arrival picks using the same set of picking parameters. Red and blue “×” marks denote the picked P- and S-wave first arrivals, respectively.

**Figure 9 sensors-26-00114-f009:**
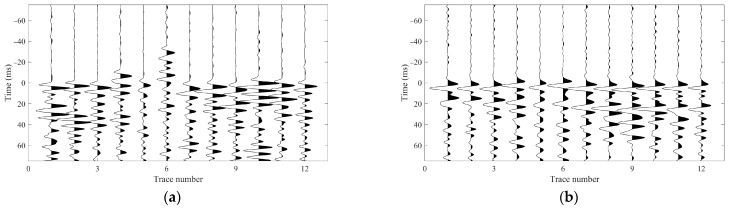
Alignment of P- and S-wave phases (flattened waveforms) based on first arrival picks obtained using two different sets of parameters for the STA/LTA energy ratio method. Panels (**a**,**b**) show the aligned P- and S-wave phases, respectively, using the first parameter set; panels (**c**,**d**) show the corresponding alignments using the second parameter set.

**Figure 10 sensors-26-00114-f010:**
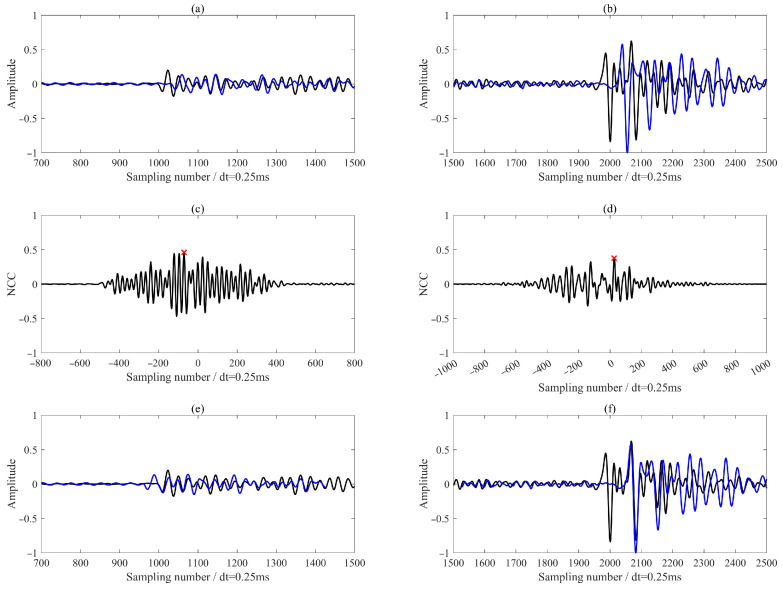
A detailed comparison of recordings from two receivers (12th and 7th) for the event shown in Figure 2a, analyzing their cross-correlation functions and the resulting time delay estimation. Panels (**a**,**b**) show the P- and S-wave phases, respectively. Panels (**c**,**d**) display the corresponding normalized cross-correlation functions, with red “×” marks indicating the locations of the maximum correlation values. Panels (**e**,**f**) show the waveforms after applying the estimated time shifts for alignment. Black and blue traces representing the recordings from the 12th and 7th receivers, respectively.

**Figure 11 sensors-26-00114-f011:**
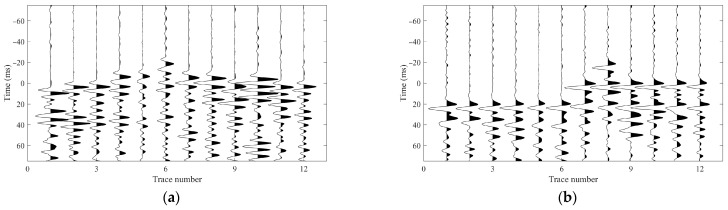
Alignment of P- and S-wave phases (flattened waveforms) based on the cross-correlation method. Panels (**a**,**b**) show the aligned P- and S-wave phases, respectively.

**Figure 12 sensors-26-00114-f012:**
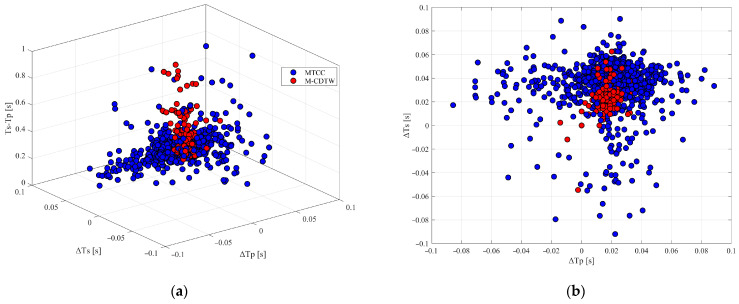
Differential travel time distributions obtained from the MTCC (blue) and proposed M-CDTW (red) methods. (**a**) Three types of differential times; (**b**) P-wave and S-wave travel time differences between the first and last receivers; (**c**,**d**) combined view illustrating inter-receiver and P-S differential time relationships.

**Figure 13 sensors-26-00114-f013:**
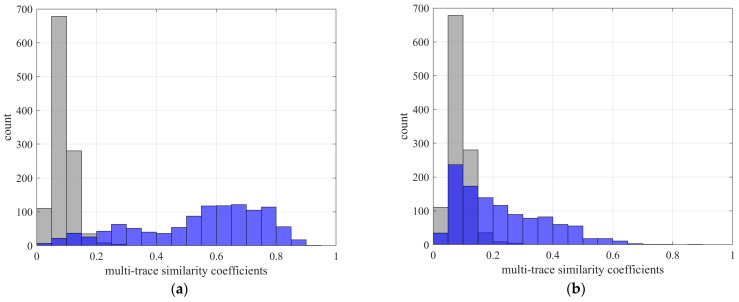
Multi-trace similarity coefficients computed after arrival time correction for P- and S-wave phases. (**a**) P-wave similarity coefficients; (**b**) S-wave similarity coefficients. Gray and light blue denote the results obtained using the conventional MTCC method and the proposed M-CDTW method, respectively.

**Figure 14 sensors-26-00114-f014:**
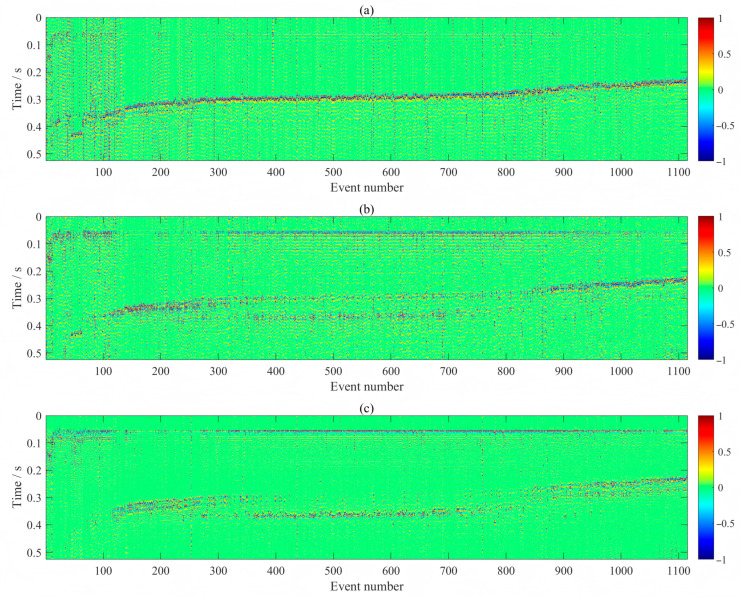
Microseismic event records extracted using P-wave arrivals picked by the proposed method, sorted in descending order of P-S time differences. Amplitudes are normalized for display. Panels (**a**–**c**) correspond to the vertical (Z) and horizontal (X, Y) components, respectively.

## Data Availability

The data and code presented in this study are available on request from the corresponding author.

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
