# Peer review of "Robust Phase Association and Simultaneous Arrival Picking for Downhole Microseismic Data Using Constrained Dynamic Time Warping"

_sensors, 2025, doi:10.3390/s26010114_

Round 1

Reviewer 1 Report

Comments and Suggestions for Authors

(1)The Introduction presents the challenges of microseismic arrival picking in overly general terms. The authors are encouraged to explicitly center the motivation around the two specific physical phenomena this work aims to address: (i) contamination of S-wave onsets by P-wave coda energy, and (ii) ambiguity in first-arrival identification caused by shear-wave splitting. These challenges should be clearly highlighted, supported by relevant literature demonstrating their prevalence and impact in downhole monitoring scenarios.
(2)The manuscript mentions “energy balancing to ensure comparability across receivers” but omits the specific algorithm used. Please clarify the implementation.
(3)Section 4 provides qualitative comparisons but would be significantly strengthened by a quantitative analysis. Incorporating objective numerical metrics to compare the proposed method with STA/LTA and cross-correlation methods is essential for a rigorous evaluation.
(4)The paper emphasizes the advantages of M-CDTW but does not address its limitations. A “Limitations and Future Work” subsection is required to discuss the method’s performance in critical scenarios, such as when the reference trace has a very low signal-to-noise ratio (SNR) or during periods of high event density with overlapping waveforms.

Author Response

Comment 1: The Introduction presents the challenges of microseismic arrival picking in overly general terms. The authors are encouraged to explicitly center the motivation around the two specific physical phenomena this work aims to address: (i) contamination of S-wave onsets by P-wave coda energy, and (ii) ambiguity in first-arrival identification caused by shear-wave splitting. These challenges should be clearly highlighted, supported by relevant literature demonstrating their prevalence and impact in downhole monitoring scenarios.

Response 1:We sincerely thank you for this insightful suggestion. In response, we have thoroughly revised the Introduction to foreground these two critical physical challenges as the core motivation of our study. A new paragraph (lines 85-102 in the revised manuscript) now explicitly articulates how (i) P-wave coda energy can obscure S-wave arrival analysis and (ii) shear-wave splitting leads to multiple, closely spaced arrivals that complicate unambiguous arrival identification. We have also incorporated key references to substantiate the prevalence and operational significance of these issues in downhole microseismic monitoring.

Comment 2: The manuscript mentions energy balancing to ensure comparability across receivers but omits the specific algorithm used. Please clarify the implementation.

Response 2:Thank you for highlighting this ambiguity. We have replaced the vague phrase “energy balancing” with a precise technical description in Section 2.4 (Step 1).

Comment 3: Section 4 provides qualitative comparisons but would be significantly strengthened by a quantitative analysis. Incorporating objective numerical metrics to compare the proposed method with STA/LTA and cross-correlation methods is essential for a rigorous evaluation.

Response 3:We agree with the reviewer on the importance of quantitative validation. Accordingly, we have expanded the performance evaluation in Section 3 (now titled “Performance Evaluation”) to include systematic benchmarking against both STA/LTA and cross-correlation methods. To systematically evaluate the reliability of the proposed method, we applied it to direct P- and S-wave first-arrival picking for all microseismic events identified from monitoring data across 19 hydraulic fracturing stages. The performance of each method was quantitatively evaluated using two key metrics in the Section 4.

Comment 4: The paper emphasizes the advantages of M-CDTW but does not address its limitations. A “Limitations and Future Work” subsection is required to discuss the methods performance in critical scenarios, such as when the reference trace has a very low signal-to-noise ratio (SNR) or during periods of high event density with overlapping waveforms.

Response 4:This is an excellent and important point. In response, we have added a dedicated paragraph titled “Limitations and Future Work” in the Discussion section (Section 4, lines 510-524).

Reviewer 2 Report

Comments and Suggestions for Authors

In this paper, a method based on nonlinear time warping during the joint analysis of two signals is proposed for the classical computational seismology problem of determining the arrival times of P- and S-waves in waveforms. This method ensures maximum waveform similarity after time transformation, using nonlinear time warping. This method, Multivariate Dynamic Time Warping, was proposed for analyzing speech fragments and microeconomic data, which exhibit significant variability in behavior on local time scales while maintaining common properties on large time scales. Regarding the application of these methods to seismic trace processing in seismic sounding applications, the question arises of their effectiveness compared to more traditional procedures for identifying the arrivals of various seismic wave types, which have long been developed and become somewhat "canonical." This issue is addressed in this paper, but the comparison is made only for the simplest STA/LTA method, and for a variant without preliminary frequency filtering and selection of the optimal frequency band. Apparently, the authors are aware of other, more sophisticated methods for detecting arrivals, for example, from reference [12] in the bibliography. Such methods include, for example, the AIC algorithm and kurtosis and wavelet-based methods. I suggest the authors conduct further research into the effectiveness of the Dynamic Time Warping method they proposed for practical application, not limiting themselves to the simplest version of the STA/LTA method.

Author Response

Comment 1: In this paper, a method based on nonlinear time warping during the joint analysis of two signals is proposed for the classical computational seismology problem of determining the arrival times of P- and S-waves in waveforms. This method ensures maximum waveform similarity after time transformation, using nonlinear time warping. This method, Multivariate Dynamic Time Warping, was proposed for analyzing speech fragments and microeconomic data, which exhibit significant variability in behavior on local time scales while maintaining common properties on large time scales. Regarding the application of these methods to seismic trace processing in seismic sounding applications, the question arises of their effectiveness compared to more traditional procedures for identifying the arrivals of various seismic wave types, which have long been developed and become somewhat "canonical." This issue is addressed in this paper, but the comparison is made only for the simplest STA/LTA method, and for a variant without preliminary frequency filtering and selection of the optimal frequency band. Apparently, the authors are aware of other, more sophisticated methods for detecting arrivals, for example, from reference [12] in the bibliography. Such methods include, for example, the AIC algorithm and kurtosis and wavelet-based methods. I suggest the authors conduct further research into the effectiveness of the Dynamic Time Warping method they proposed for practical application, not limiting themselves to the simplest version of the STA/LTA method.

Response 1:We sincerely appreciate your insightful suggestions. We have restructured the paper to better highlight the methodological motivation and validation. Specifically, Section 3 now focuses on two representative microseismic events to systematically analyze the limitations of conventional picking approaches, particularly the STA/LTA and waveform cross-correlation methods, and elucidate the underlying causes, such as P-wave coda interference and shear-wave splitting.

In the Discussion section, we extend the evaluation to a larger dataset comprising 1,115 microseismic events identified from field monitoring records. By comparing our M-CDTW results against those obtained using the MTCC method, we demonstrate that MTCC often fails to deliver consistent and physically plausible picks when confronted with complex real-world waveforms. To quantitatively assess performance, we employ two complementary metrics: (1) the compactness of inter-receiver differential travel-time distributions, and (2) multi-trace similarity coefficients computed after arrival-time correction. The significantly tighter clustering of time differences and higher waveform coherence achieved by our method underscore its superior robustness and reliability in practical downhole microseismic monitoring scenarios.

Reviewer 3 Report

Comments and Suggestions for Authors

The article submitted for review has undoubted practical value and, after answering the questions raised, can be published in the Sensors journal.

Author Response

Comments 1: The precision of microseismic source location and parameter estimation critically hinges on accurate phase association and arrival time picking». What is meant by the term "accurate phase association"? Specify. This is the nature of the wave (P, S, Surface, etc.), or (type of wave: direct, reflected, refracted, etc.).

Response 1: We apologize for the lack of clarity. In the revised Introduction (Section 1), we have provided a clear and specific definition: “The accuracy of microseismic source location, focal mechanism estimation, and the subsurface velocity model reconstruction critically depends on two interdependent tasks: precise phase association (e.g., direct P-waves, direct S-waves, SH and SV waves from shear-wave splitting, and occasionally reflected and refracted phases) and high-fidelity arrival time picking [9-12].”

Comments 2: While performance can be enhanced through pre-processing (e.g., denoising) or the integration of multi-attribute constraints, the Signal-to-Noise Ratio (SNR) remains the primary limiting factor governing picking reliability [16]». Explain what is "the integration of multi-attribute constraints"?

Response 2: We have clarified this term in the Introduction (Section 1) by providing concrete examples: “...or the integration of multi-attribute constraints (such as, joint optimization of MER and the Akaike Information Criterion (AIC), or hybrid frameworks combining STA/LTA, polarization analysis, and AIC) [16, 21]”.

Comments 3: "traditional methods often yield unsatisfying results". What methods are we talking about?

Response 3: We have specified the methods referred to as “traditional” in the Introduction (Section 1): “Existing methodologies often struggle to reliably identify true first arrivals under complex field waveforms, such as near-source structural complexity, low inter-trace waveform coherence, and interference from overlapping seismic phases. This challenge is particularly pronounced in specific downhole monitoring configurations, where conventional picking methods, such as time-windowed attribute analysis employing fixed thresholds or waveform cross-correlation for inter-trace time-delay estimation under conditions of low waveform coherence, often yield inconsistent or erroneous picks, even for events exhibiting relatively high amplitudes.”

Comments 4: These effects generate highly complex wave- fields in which P- and S-phase arrivals are frequently obscured or distorted [17]». Yes, indeed S-waves can be distorted by these effects, but why does this happen with direct P-waves? Please explain.

Response 4: The original phrasing was imprecise. We have corrected it. “Typical manifestations include P-wave coda energy contaminating the S-wave onset picking window and near-surface reflected phases arriving immediately after direct arrivals [22]. Consequently, conventional pickers struggle to reliably identify true onsets across a wide range of P/S energy ratios, ultimately introducing substantial errors into event location and source characterization [23]”.

Comments 5: To address these specific challenges, P-wave coda contamination and shear-wave splitting, this paper proposes a novel multi-channel arrival picking framework based on Constrained Dynamic Time Warping (C-DTW). The name of the technique "Constrained Dynamic Time Warping" sounds like some kind of fantastic process or a term from Einstein's special theory of relativity. Maybe the authors should reconsider the name of the technique, for example: Limited dynamic distortion of the time interval.

Response 5: We appreciate the reviewer's creative suggestion. However, "Constrained Dynamic Time Warping (CDTW)" is a well-established and widely recognized term in the fields of time-series analysis and pattern recognition, specifically referring to DTW with a Sakoe-Chiba band or similar global constraint. Changing the name would create confusion and disconnect our work from the existing literature. We have therefore retained the standard nomenclature.

Comments 6: DTW aligns time series by identifying an optimal warping path that minimizes the cumulative distance between two sequences». Please explain this phrase. What is "an optimal warping path" and what is "the cumulative distance between two sequences".

Response 6: We have expanded the explanation in Section 2.1 to make these concepts more accessible.

Comments 7: Our approach capitalizes on two key physical priors: (1) waveform similarity across the receiver array for a common source, and (2) spatial consistency in inter-receiver travel-time differences governed by the local velocity structure». However, the waveform of a signal from the same source can vary significantly on recordings from different receivers located at different epicentral distances and at different propagation azimuths. This is a well-known fact. The second part of the phrase requires explanation. What is "spatial consistency in inter-receiver travel-time differences", clear explanations are needed.

Response 7: We have clarified this physical prior in the Introduction (Section 1) and Section 2.4 in the revised manuscript.

Comments 8: The final DTW distance is D(n,m), and the optimal path is recovered by backtracking from (n,m) to (1,1). This warping path that minimizes the total cumulative distance between the Z-components of two microseismic signals, is illustrated in Figure 1». But Figure 1 shows almost identical microseismic signals along the vertical and horizontal axes, so it's enough to simply perform the procedure "that minimizes the total cumulative distance between the Z-components of two microseismic signals". Show two different signals along these axes that propagated at different distances with different speeds. What does the color scale in Figure 1 mean?.

The same situation is typical for Figure 2, where two almost identical waveforms are compared. Please provide the results of the DTW algorithm for two different microseismic signals that are obtained at epicentral distances that differ from each other by 2 or more times.

Response 8: We understand the reviewer's request. The figures in the manuscript use real field data from closely spaced receivers (10m), so their similarity is realistic. The purpose of Figure 1 (now Figure 3a in the revised manuscript) is to illustrate the concept of the warping path. We have added a caption note: “The white line represents the optimal warping path; the background color map (not shown in this schematic) would represent the local cost d(i,j), where darker colors indicate a smaller distance (better match).” Furthermore, Figure 3b in the revised manuscript now shows a more compelling example of alignment between receiver #1 and 12, which are 110 m apart, demonstrating the method’s ability to handle more significant differences.

Comments 9: Commenting on section 3. “Field Data Tests”, it is necessary to ask the question: why do the authors not use the S phase allocation method using the Vadati graph in their algorithms? The reliability of determinations of arrival times can be controlled using the Wadati diagram which demonstrates a linear relationship between the arrival times of P-waves from the source to a station and the differences S–P in the travel times of S- and P-waves for a set of stations:

ts-tp=(1-Vp/VS)(tp-t0)

where tP and tS are P- and S-wave arrival times; VP and VS are the P- and S-wave velocities, respectively; and t0 is the origin time (Kovachev S. A., and Krylov A. A. Results of Seismological Monitoring in the Baltic Sea and Western Part of the Kaliningrad Oblast Using Bottom Seismographs // Izvestiya, Physics of the Solid Earth, 2023, Vol. 59, No. 2, pp. 190–208).

Response 9: Thank you for your kind suggestion. While the Wadati diagram is indeed a powerful tool for evaluating P- and S-wave arrival times across a spatially distributed seismic network, it is not directly applicable to the problem addressed in this study. Our work focuses on the single-event, multi-channel arrival picking task: the accurate identification of first arrivals from noisy and complex waveforms recorded by a closely spaced downhole geophone array. However, the underlying assumptions of the Wadati method are typically violated in the near-source, highly heterogeneous environment characteristic of hydraulic fracturing monitoring, rendering it unsuitable for our setting.

To rigorously quantify the performance of our proposed M-CDTW framework relative to conventional approaches (e.g., STA/LTA and cross-correlation), we introduce two objective, waveform-based evaluation metrics, which are presented in Figures 12-14 of the revised manuscript:

(1) Inter-receiver differential travel-time concentration: For each event, we compute the distribution of inter-receiver travel-time differences, for P-waves and S-waves separately, across all receiver pairs. More accurate and consistent picks yield narrower, more concentrated distributions (i.e., lower variance), reflecting physically plausible and spatially smooth travel-time fields.

(2) Multi-trace similarity coefficient after alignment: Following arrival-time correction, (i.e., shifting each trace to a common reference time based on the picked arrivals),we compute the average zero-lag normalized cross-correlation (NCC) among all traces within a fixed window centered on the aligned phase. A higher average NCC indicates greater waveform coherence and, consequently, more precise and robust arrival picks.

These metrics provide a quantitative basis for assessing picking consistency and waveform fidelity without reliance on truth arrival times or external constraints, thereby enabling a self-contained evaluation framework specifically tailored to downhole arrays.

Round 2

Reviewer 2 Report

Comments and Suggestions for Authors

The authors made a big work and took into consideration all remarks. I suppose that now the paper is suitable for publishing in present form.

Author Response

Comments 1: The authors made a big work and took into consideration all remarks. I suppose that now the paper is suitable for publishing in present form.

Response 1: Thank you very much for your thoughtful comments and positive assessment. Your constructive feedback has significantly enhanced the quality and clarity of our manuscript.

Reviewer 3 Report

Comments and Suggestions for Authors

The authors of this paper have taken all my comments into account in the second version and provided extensive clarifications in their response to the reviewer. This version of the article may be published in the journal Sensors.

Author Response

Comments 1: The authors of this paper have taken all my comments into account in the second version and provided extensive clarifications in their response to the reviewer. This version of the article may be published in the journal Sensors.

Response 1: Thank you very much for your thorough review and kind acknowledgment of our revisions. We sincerely appreciate your constructive feedback, which greatly helped us improve the manuscript.